## Overview Review

refugee; forced displacement; social support; forced migration; mental health

**Corresponding author:**
Gulsah Kurt;
Email: g.kurt@unsw.edu.au

# Social support coping strategies among sub-Saharan African refugees: A systematic review and meta-synthesis

Tilahun Kassew Gebeyehu[1,2] , Ruth Wells[1], Max Loomes[1], Zachary Steel[1] and Gulsah Kurt[3]

[1]Discipline of Psychiatry and Mental Health, UNSW, Australia; [2]Psychiatry, University of Gondar, Ethiopia and [3]School of Psychology, UNSW, Australia

## Abstract

Social support is a key coping resource; its specific role for refugees from sub-Saharan Africa in high-income settings remains poorly understood. This systematic review synthesises existing evidence on the sources of social support and how these networks aid coping during resettlement. We applied an optimised search strategy to identify studies examining social support among sub-Saharan African refugees across six academic indexing databases. We then undertook a meta-synthesis of the identified studies. This involved the use of meta-thematic analysis of the interpretations and quotes presented in each study, combining thematic analysis through the reviewer's reflexivity. The PRISMA framework guided the review process to ensure methodological rigour. A total of 22 articles were included in the qualitative meta-synthesis. The synthesis revealed four key sources of social support: 1) family, 2) friends, 3) ethnic and community groups, and 4) cultural and religious supports. These support sources played multiple roles, including enhancing community engagement and reciprocity, providing practical and emotional assistance, offering relief from distress and cultivating cultural continuity and adaptation. However, some individuals distanced themselves from their ethnic community and preferred self-driven coping. Access to social support systems remains a crucial coping resource for many sub-Saharan African refugees in high-income settings, alleviating distress and enhancing resilience. Programs that strengthen informal social support networks through community-driven initiatives can enhance the relevance of social support. Future research should investigate the role of social support across various phases of resettlement in relation to psychosocial well-being.

## Impact statement

The Sub-Saharan Africa region has experienced decades of violence, conflict and political upheavals. Refugees from this region who have resettled in high-income countries often struggle with conflict-related trauma and displacement-related stressors. Social support plays a key role in their coping with these stressors. However, to date, no study has systematically reviewed the existing literature on the roles and sources of social support in facilitating the coping process among those resettled in high-income countries. Our review identified 22 studies on social support and coping among refugees from Sub-Saharan Africa resettled in high-income countries (e.g., Australia, Canada, Europe and the United States of America). Family, friends, community and cultural and religious supports emerged as key sources of social support. Social support played multiple roles in the coping process of those individuals by providing a sense of belonging, promoting reciprocity, alleviating distress and supporting adaptation to a new life. Despite these benefits, some studies also identified potential risks associated with social support. Together, our findings provide a comprehensive review of social support as an inherent coping mechanism for Sub-Saharan African refugees in high-income countries and offer an evidence base to inform culturally responsive psychosocial interventions and services.

## Introduction

Conflict-related trauma has profound effects on individuals, families and communities, often leading to severe psychological distress and long-term mental health challenges (Blackmore et al., 2020; Mesa-Vieira et al., 2022). The sub-Saharan African region has endured decades of armed conflict, civil war and political instability, resulting in the displacement of millions of people (Epule et al., 2015; Merry et al., 2017). Many forcibly displaced persons from this region have

initially sought temporary safety in neighbouring countries (Salehyan, 2014), while others have been resettled in more distant, high-income countries. Cultural, social and religious differences often complicate their resettlement experience (Corin, 2017; Ward et al., 2020).

Many sub-Saharan African societies emphasise communal orientations in which individuals are embedded within inter-dependent networks of kinship, community and spiritual bonds, in contrast to the more individualistic perspectives common in many industrialised Western societies (Onunwa, 1994; Oludare, 2015). Despite cultural and linguistic diversity across the Sub-Saharan region, social relationships remain embedded in every-day life and social organisation (Elliesie, 2017; Na et al., 2019). Within these societies, social support networks such as family, neighbours, tribes, ethnic groups and religious communities pro-vide not only emotional and practical support but also a strong sense of meaning, belonging and continuity during times of hardship (Makiwane and Kaunda, 2018; Adeyeye, 2024). In times of crisis, these networks become central, mobilising collective resources and shared values to support individuals in need. Although conflict and forced displacement profoundly disrupt these social networks and resources, individuals often find ways to restore, adapt or build new ones that continue to play an import-ant role in coping with displacement-related stressors (Crisp, 2010; Bamidele and Pikirayi, 2024).

In low-resource settings or refugee camps, many traditional community structures are rebuilt and become pivotal in providing essential services and assistance to forcibly displaced people, as they are often densely settled (Miller and Rasmussen, 2010). However, this pattern can differ in high-income resettlement contexts, where family and community structures may be sub-stantially disrupted by restrictive resettlement policies, such as dispersal that might limit ethnocentric resettlement patterns (Van Liempt, 2011; Stewart and Shaffer, 2015; Ermansons et al., 2023). While these policies aim to reduce pressure on major urban centres by balancing costs, ensuring proportional access to hous-ing and enhancing opportunities for integration, they might also limit opportunities to establish cultural and communal networks. As a result, refugees often experience substantial disruptions to their traditional sources of support rooted in culture and com-munity (Cernea and McDowell, 2000). These challenges are fur-ther compounded by acculturative stress arising from language barriers, unfamiliar cultural contexts and exposure to Western individualist values that may contradict collectivistic values (Ritblatt and Hokoda, 2022).

Given the centrality of social relationships in many Sub-Saharan African societies, the ways in which social support is accessed and utilised may differ in high-income settings from those in other contexts. Despite the importance of family- and community-based support systems, the academic literature remains unclear about the extent to which these forms of social coping can be utilised during displacement and resettlement. Although some studies have explored the role of social support among resettled sub-Saharan African refugees (Goodman, 2004; Grupp et al., 2022), no study has systematically synthesised the evidence on the role of social support and networks in managing stress among those resettled in high-income contexts. To address this gap, this study aimed to systematically review existing studies on social support among Sub-Saharan African refugees who resettled in high-income countries, with the goal of developing a comprehensive understanding of how social support contributes to coping with displacement and resettlement challenges. Therefore, this systematic review addressed the following research questions:

1. What are the common sources of social support used by Sub-Saharan African refugees resettled in high-income countries?
2. How does social support help those individuals to cope with stress in these settings?

## Methods

### Study design

A systematic review and qualitative meta-synthesis were conducted following the Preferred Reporting Items for Systematic Review and Meta-Analysis (PRISMA) (Page et al., 2021). The compiled PRISMA checklist table is provided in the Supplementary Material. The review protocol was developed and registered in the Prospero database (Reference Number: CRD42023451868). This systematic review focuses on one of the aims of the registered review: coping strategies among forcibly displaced people from Sub-Saharan Africa in high-income settings.

### Data searching strategy

We applied an optimised search strategy following the methods established by Steel et al. (Steel et al., 2009; Steel et al., 2014). This involved identifying an initial pool of 12 articles that met the inclusion criteria of the current review. This set of articles was then used to identify and iteratively refine the search terms and search strategy to maximise sensitivity and specificity in identifying rele-vant articles. The resulting search strategies achieved a sensitivity of 83.3% or above against the initial relevant articles. We searched six databases, including PubMed, Scopus, PsycINFO, Embase, Web of Science and Cumulative Index to Nursing and Allied Health (CINAHL), using tailored keywords. Additional manual searches on Google Scholar were conducted to identify the reference lists of relevant articles. The search process was developed in PubMed, and iterative changes were made for other databases, for instance, using major headings (MH) before the subject headings in CINAHL, rather than using medical subject headings (MeSH) in PubMed. The search was conducted using combinations of five key subject headings (MeSH) terms and keywords from titles and abstracts.

The key concepts were combined using Boolean operators (AND/OR). For instance, the key concept "Coping strategy" was searched for using "OR" Boolean operator to combine subject headings with keywords: *"Adaptation, psychological [MeSH] OR "Coping strategies" OR "coping behaviour" OR "coping skills" OR "coping mechanisms" OR coping OR cope\* OR adapt\* OR "Psycho-logical Coping" OR "mental health."* Then the five key concepts search was combined using the "AND" Boolean operator: *Coping AND Armed conflict AND Sub-Saharan African AND Refugees AND Developed countries.* During the search process, the keywords were optimised to retrieve more relevant articles, providing a more sensitive search while reducing the specificity of the results.

The search was restricted to studies published between 1990 and April 2025 to capture the relevant research conducted over the past three decades. The 1990s marked a significant period of increased displacement of people from sub-Saharan Africa due to conflict (Adepoju, 1995; Akokpari, 1998; Ammar and Nohra, 2014), leading to a rise in research on refugee mental health and coping resources (Abidde, 2021). The initial search of each database was conducted

on 15 October 2023, and the results were updated for publication on 16 June 2025.

## Eligibility criteria

### Inclusion criteria

The studies meeting the following criteria were included in the review if they:

1. Were qualitative or mixed-method (qualitative and quantitative) studies.
2. Included a sample of resettled sub-Saharan African refugees, defined as being equal to or above 50%.
3. Explored "coping strategies," "coping behaviour," or "social support coping."

### Exclusion criteria

Studies were excluded if they focused solely on Sub-Saharan Africans forcibly displaced within the African continent or in low- and middle-income settings, or if they primarily included Sub-Saharan Africans who migrated to high-income settings for educational or employment purposes, rather than due to forced displacement. Studies with unclear outcomes of interest or consisting solely of case reports were also excluded from the current analysis (see Figure 1).

### Screening of the studies

We used Covidence software to facilitate data screening. Covidence is a web-based software program developed to manage the various stages of the systematic review screening process. Duplicates not identified by the software were removed manually

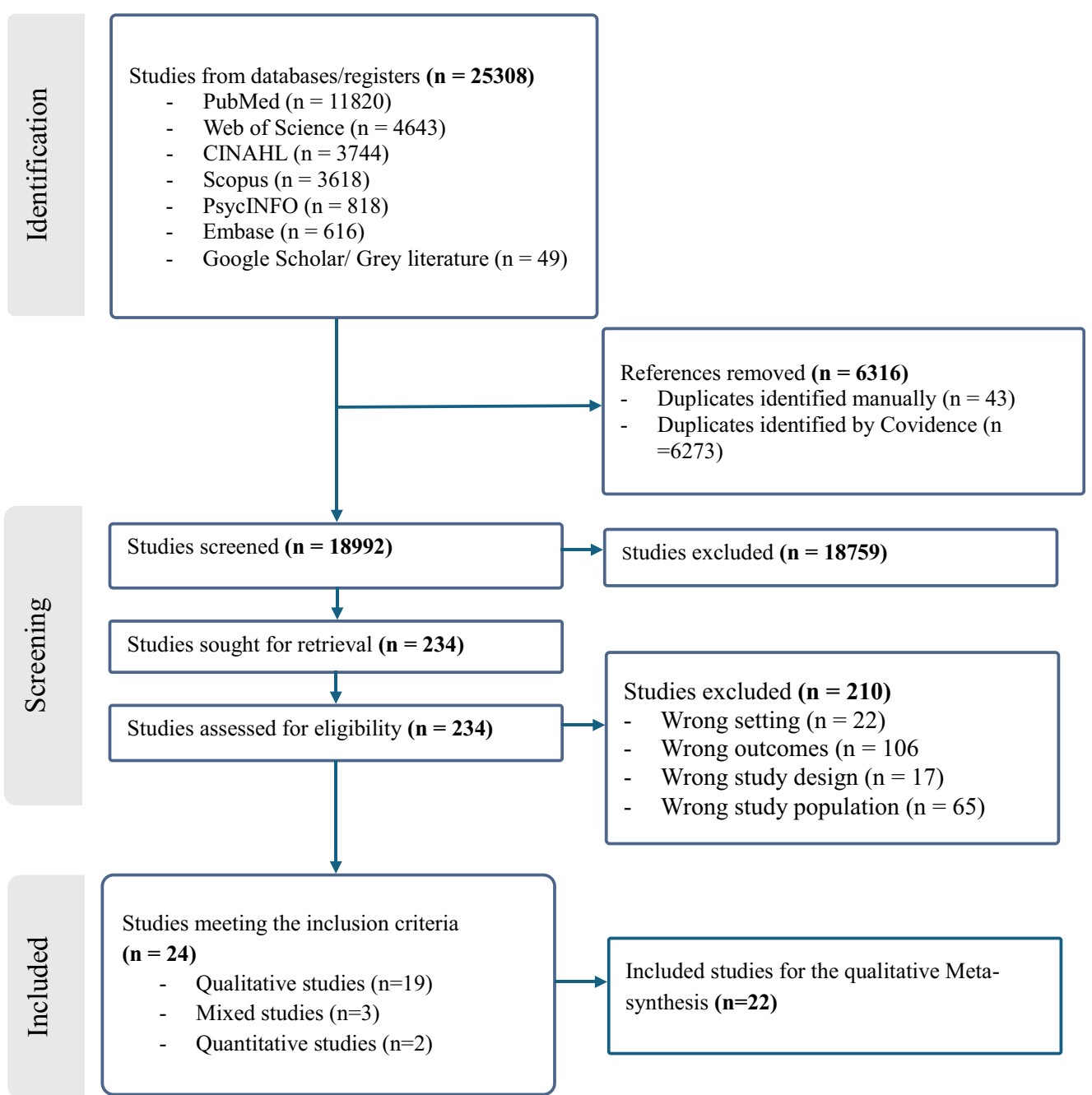

**Figure 1.** PRISMA 2020 Flow diagram studies screening and inclusion and exclusion process.

during screening. The first author and author-3 (ML) screened the titles, abstracts and full texts of the identified papers. After the full-text screening, a data extraction template was uploaded to Covidence, enabling data extraction within the software. The references for the final set of included articles are presented in Table 1. A PRISMA flow diagram documenting the review stages (Figure 1).

### Data extraction

The required data for the articles were extracted using a structured, customised data abstraction spreadsheet. This spreadsheet included the author's name, publication year, study aims, study design, sampling technique, sample size, data collection technique, instrument, data analysis, gender, age, home country, resettlement country, length of stay in resettlement, main findings (such as types of coping), author interpretations and participant quotes. Data were extracted by the first author using both Covidence and Microsoft Excel. Software-extracted data were exported to an Excel spreadsheet for cross-checking consistency. All relevant author interpretations and participant quotes from the included articles were collected on a separate Microsoft Excel spreadsheet. Finally, the Excel spreadsheet data was used to analyse and synthesise the review and qualitative meta-synthesis.

### Quality assessment

We used the Mixed Methods Appraisal Tool (MMAT) to assess the methodological quality of the included studies (Hong et al., 2018). The MMAT tool consists of seven questions for each study design and article. The responses to the appraisal questions are "Yes," "No," or "Cannot tell." The first two questions are common for the five categories of study methods. Each article was considered for further appraisal if it responded "Yes" to the first two questions. As per the tool, each included study was rated by selecting the appropriate study category questions. For each evaluated study, the overall quality score is presented utilising characters such as stars (*) or % (see Table 3). The first author independently assessed the quality appraisal, then reviewed the quality ratings for the 11 articles with authors RW and GK to cross-check and reach consensus.

### Data synthesis and analysis

Descriptive statistics were used to summarise and depict the key characteristics and findings of the included studies. We then applied a narrative synthesis to summarise texts and tables. While a few studies employed quantitative or mixed methods designs, the review primarily focused on qualitative findings. Quantitative reports were reviewed for contextual relevance but were not included in the thematic meta-synthesis. The analytical approach drew on constructivist epistemological perspectives (Pouliot, 2007; Burr et al., 2019), emphasising the recursive and reflexive processes between the primary researcher's lived experience and studies reviewed (Breckenridge et al., 2012). This approach recognises that the findings reported herein constitute a situated interpretation socially and contextually constructed through the researcher's positionality, perspectives and engagement with participants' accounts, as documented in these 22 research papers.

### Positionality reflexivity

The first author, who comes from a sub-Saharan African background, has worked closely with Ethiopian and Somali forcibly displaced people in refugee camps in Ethiopia as a mental health and psychosocial support professional. These working experiences have led the first author to observe the day-to-day lived realities of forcibly displaced people, including their struggles with loss, trauma and adverse life events. Despite these difficulties, the first author has witnessed the strengths of many sub-Saharan African refugees, particularly the significant role that family support and religious beliefs play in their lives. At the same time, the first author's professional background enables him to approach the different social, cultural and religious narratives and experiences with empathy and cultural sensitivity. As someone who shares cultural and regional closeness with many of the study participants, the first author brings his lived experiences to help interpret meaning within the context. Furthermore, the rest of the research team brings diverse professional and research experiences with forcibly displaced people across multiple settings. Their experiences with different communities have informed their understanding of the complex socio-cultural and psychosocial issues that influence the lived experiences of forcibly displaced people. Therefore, our various positionalities as researchers were undoubtedly reflected in how we engaged with the research, analysed the data and interpreted the findings.

### Qualitative thematic meta-synthesis

This thematic meta-synthesis process followed a three-stage approach (Thomas and Harden, 2008; Caskurlu et al., 2021).

1. The free-line-by-line coding of the findings from primary studies: The text was extracted from the findings of the included studies, one row at a time, onto a separate Excel spreadsheet. Memos and codes were created inductively to capture the meaning and content of each sentence in the illustrations and participant quotes. The codes emerged from the data itself based on the meaning contained in each sentence of the text. These codes were compared across studies to translate concepts and identify commonalities and differences. All text with a given code was reviewed to assess consistency of interpretation and determine whether additional levels of coding were necessary. This process resulted in 46 codes after re-reading and further refining the data. A rich, nuanced understanding of the data across the studies ensured that the codes reflected the diverse perspectives of the included studies.
2. The organisation of the "free codes" into related areas to construct "descriptive" themes: Codes were grouped in a hierarchical tree structure, looking for similarities and differences. The new codes were created to capture the meaning of groups of initial codes
3. The development of "analytical" themes: This stage of qualitative synthesis is the equivalent of "third-order interpretation" in meta-ethnography for the development of themes that go beyond the content of the primary studies (Thomas and Harden, 2008). This was achieved based on the descriptive theme that emerged from the inductive synthesis of the primary study findings. By capturing the descriptive themes, the abstract and analytical themes began to emerge. The developed themes were re-examined, and changes were made as necessary in discussions with the co-authors. This process yielded four overarching themes: the sources of social support, the reasons for seeking social support, the role of social support and the limits of social support.

**Table 1.** Included qualitative research studies and key study details

| | References | Study design & sample (n) | Age (mean/range) | Gender | Sampling | Data analysis | Data collection technique | Home country | Resettlement country | Length of resettlement |
|---|---|---|---|---|---|---|---|---|---|---|
| [1] | Goodman (2004) | Qualitative (n = 14) | 16–24 years | M = 14 | Case-centered sampling | Narrative analysis | Face-to-face interview | South Sudan | USA | 6–12 months |
| [2] | Schweitzer et al. (2007) | Qualitative (n = 13) | M = 29.8 years | F = 4 M = 9 | Convenience sampling | Interpretative Phenomenological Analysis | Face-to-face interview | South Sudan | Australia | Mean = 4.15 years |
| [3] | Khawaja et al. (2008) | Qualitative (n = 23) | 35 years | F = 12 M = 11 | Snowball sampling | Interpretative phenomenological analysis | Semi-structured questions for an In-depth interview | S. Sudan | Australia | Mean = 2.55 years |
| [4] | Murray (2010) | Mixed method (Quantitative: n = 90 Qualitative n = 10) | Qual m = 36.5 years Quan m = 34.2 years | F = 56 M = 34 Qualitative: F = 5 M = 5 | Snowball sampling | Hierarchical regression and Miles and systematic Huberman analysis | Face-to-face interview | South Sudan | Australia | Mean = 3.45 years, SD = 2.1 |
| [5] | Shakespeare-Finch and Wickham (2010) | Qualitative (n = 12) | 19–60 years | F = 6 M = 6 | Convenience sampling | Interpretative Phenomenological Analysis | Face-to-face interview | South Sudan | Australia | <=5 years |
| [6] | Puvimanasinghe et al. (2014) | Qualitative (n = 24) | 18–56 years | F = 14 M = 10 | Snowball sampling | Narrative analysis | Individual interview | Burundi & Sierra Leone | Australia | Range: 3–10 years |
| [7] | Betancourt et al. (2015) | Qualitative (youth: n = 30; parents; n = 32) | 16–25 years & 28–59 years | F = 33 M = 29 | Purposive sampling | Grounded theory-derived content analysis | Focus group Interview | Somalia | USA | Mean Youth = 5.4 years Parents = 9.2 years |
| [8] | Markova and Sandal (2016) | Mixed (Quantitative n = 95 Qualitative n = 10) | Quali: m = 33 years. Quan m = 28 years | F = 42 M = 53 F = 6 M = 4 | Convenience sampling | Descriptive and Explanatory Analysis | Survey and focus-group interviews | Somalia | Norway | Quantitative Mean = 10 years Qualitative Mean = 3 years |
| [9] | Joyce and Liamputtong (2017) | Qualitative (n = 16) | 16–24 years | F = 7 M = 9 | Purposive sampling | Thematic analysis | In-depth interview and Photo elicitation | DR Congolese | Australia | Range = 1–10 years |
| [10] | King et al. (2017) | Qualitative (n = 15) | 21–57 years | F = 8 M = 7 | Purposive sampling | Thematic analysis | One-to-one interview & Focus group discussion Using a photovoice approach. | Ethiopia, Eritrea, South Sudan, Rwanda, DR Congo, and Sierra Leone | Canada | > = 4 years |
| [11] | Omar et al. (2017) | Qualitative (n = 36) | 18–60 years | M = 36 | Purposive sampling | Thematic analysis | Focus-group Interview | Somalia & Eritrea | Australia | Not specified |
| [12] | Abraham et al. (2018) | Qualitative (n = 18) | Range: 18–60 years | F = 18 | Purposive sampling | Content-Focused Hermeneutic Analysis | Focus group and In-depth interview | Eritrea | Norway | Range = 1–8 years |
| [13] | Covington-Ward et al. (2018) | Qualitative (n = 34) | Not specified | F = 14 M = 20 | Purposive sampling | Grounded theory approach, Systematic analysis | Focus Group Interview | 10 Sub-Saharan African countries | USA | Mean = 14.2 years |
| [14] | Vromans et al. (2018) | Qualitative (n = 10) | 22–53 years | F = 10 | Purposive sampling | Feminist framework approach analysis | Focused group discussion | DR Congo, Eritrea, Rwanda, South Sudan and Afghanistan | Australia | Mean = 2.4 years |

**Table 1.** (*Continued*)

| | References | Study design & sample (n) | Age (mean/ range) | Gender | Sampling | Data analysis | Data collection technique | Home country | Resettlement country | Length of resettlement |
|---|---|---|---|---|---|---|---|---|---|---|
| [15] | Olukotun et al. (2019) | Qualitative (n = 24) | M = 35.7 years | F = 24 | Purposive sampling | Thematic analysis | Face-to-face & phone interview | Sub-Saharan African | USA | Mean = 11.52 years |
| [16] | Woodgate and Busolo (2021) | Qualitative (n = 28) | 15–29 years (64% > 18 years) | F = 9 M = 19 | Purposive sampling | Thematic analysis | In-depth interview | DR Congo, Burundi, Somalia, Sudan | Canada | Mean = 4.87 years |
| [17] | Ikafa and Hack-Polay, 2018 | Qualitative (n = 30) | 18–59 years | F = 19 M = 11 | Convenience sampling | Thematic analysis | In-depth face-to-face interview | Sub-Saharan African countries | Australia | Range = 1–17 years |
| [18] | Goitom and Idemudia (2022) | Qualitative (n = 15) | M = 36 years | F = 7 M = 8 | Purposive sampling | Content analysis | Individual In-depth interview | Ethiopia | Canada | Not specified |
| [19] | Grupp et al. (2022) | Mixed (Quantitative = 119 Qualitative = 26) | 28 years | F = 34 M = 85 Qualitative: F = 3 M = 23 | Stratified & Snowball sampling | Chi-square and T-test vs. Thematic analysis | Face-to-face using Structured questionnaire vs. Focus group interview | Eritrea, Somalia, Cameron, Ethiopia, Nigeria, Togo & Sudasn | Germany | Mean = 2 years |
| [20] | Pittaway and Dantas (2022) | Qualitative (n = 23) | 15–21 | F = 7 M = 16 | Purposive sampling | Thematic analysis | Focus group discussion | South Sudan | Australia | Not specified |
| [21] | Scott et al. (2022) | Qualitative (n = 5) | 16–21 years | M = 5 | Convenience sampling | Interpretative Phenomenological Analysis | In-depth interview | South Sudan, Eritrea | United Kingdom | Range: 6 months to 2 years |
| [22] | DiClemente-Bosco et al. (2024) | Qualitative (n = 15) | 24–35 years | F = 14 M = 1 | Purposive sampling | Grounded theory approach | In-depth interview | Sub-Saharan Africans | USA | Range 1–10 years |

## Results

### Search results

The combined search strategy across all databases (PubMed, Web of Science, CINAHL, Scopus, PsycINFO, Embase and Google Scholar) returned 18,992 unique articles. Review of the titles and abstracts resulted in the exclusion of 18,759 articles that did not meet the inclusion or exclusion criteria, and 234 articles were identified for full-text review. Following the full-text review, 24 papers met the inclusion criteria. Of these papers, 19 applied qualitative methodologies, three applied mixed methods and two reported only quantitative findings. As the current study employed a meta-synthesis framework, we excluded the two studies that reported only quantitative findings, resulting in a final pool of 22 papers for the meta-synthesis analysis (Figure 1).

### Characteristics of included studies

The 22 articles included a total of 463 participants (*male = 243, female = 220*). All studies were conducted between 2004 and 2024 and undertaken within six high-income countries representing resettlement patterns across recent decades. The countries included Australia (Schweitzer et al., 2007; Khawaja et al., 2008; Murray, 2010; Shakespeare-Finch and Wickham, 2010; Puvimanasinghe et al., 2014; Joyce and Liamputtong, 2017; Omar et al., 2017; Ikafa and Hack-Polay, 2018; Vromans et al., 2018; Pittaway and Dantas, 2022), Canada (King et al., 2017; Woodgate and Busolo, 2021; Goitom and Idemudia, 2022), Germany (Grupp et al., 2022), Norway (Markova and Sandal, 2016; Abraham et al., 2018), United Kingdom (Scott et al., 2022) and the United States of America (USA) (Goodman, 2004; Betancourt et al., 2015; Covington-Ward et al., 2018; Olukotun et al., 2019; DiClemente-Bosco et al., 2024). Table 1 provides an overview of the participants and methodological characteristics of the included studies. Seven of the included articles reported that their participants came from four or more different sub-Saharan African countries (King et al., 2017; Covington-Ward et al., 2018; Ikafa and Hack-Polay, 2018; Vromans et al., 2018; Olukotun et al., 2019; Woodgate and Busolo, 2021; Grupp et al., 2022).

The sampling techniques used were primarily purposive and snowball sampling, and thirteen studies collected data through face-to-face, in-depth, individual interviews (Table 2).

### Quality assessment

The methodological quality of the included studies was generally acceptable, with all reports scoring 3 (3\*\*\*) or higher on the MMAT criteria. Thirteen studies received a maximum score, and six met 4 of the 5 MMAT quality criteria. All the studies presented straightforward research questions to undertake the study. The research methodology employed in most studies was appropriate for answering the research questions (Table 3).

### The qualitative meta-synthesis

Data extraction from the 22 studies identified 164 quotes, reflecting the core themes and sub-themes extracted across the reviewed papers. This mapped to 45 codes aligned with themes and constructs identified in the original papers. The meta-synthesis of the original quotes, descriptions and mapped codes identified four emergent themes in the findings reported across the reviewed studies: the key sources of social support, the reasons for seeking

social support, the role of social support and the limits of social support.

### Theme 1: Sources of social support

Social support consistently featured as a central protective factor in the interviewed participants' efforts to cope with trauma, displacement and the challenges of resettlement across 22 studies. Participants described how the main sources of social support came from community networks to which they were connected, as well as from family members, friends and cultural or religious groups. These forms of support were identified not only as providing emotional reassurance but also practical assistance and a sense of identity and belonging. The sources of support identified operated across both formal and informal domains and were often deeply shaped by cultural traditions and shared migration experiences. The theme is presented through four interrelated sub-themes: community support [2, 5, 6, 8, 11, 12, 13, 14, 17, 18, 19, 20, 21, 22], family support [1, 4, 7, 8, 9, 15, 16, 17], friends support [1, 2, 3, 4, 5, 6, 8, 9, 15, 17, 21] and cultural and religious support [6, 7, 8, 9, 10, 11,14, 18, 19] (Figure 2).

Community support. Access to community support was identified as an important source of resources and coping in navigating past trauma and post-migration stressors in the quotes reported within 17 of the primary studies presented in nine codes. The

**Table 2.** Methodological characteristics of qualitative reports (*n* = 22)

| Study characteristics | | Reports *n* (%) |
|---|---|---|
| Sex of participants | Female | 220 (47.5%) |
| | Male | 243 (52.5%) |
| Participant age range | Range | 15–60 years |
| Sampling technique | Purposive | 12 (54.5%) |
| | Snowball | 4 (18.2%) |
| | Convenience | 5 (22.7%) |
| | Case centred | 1 (4.6%) |
| Data collection | Individual/ in-depth interview | 13 (59.1%) |
| | Focus group discussion | 7 (31.8%) |
| | Combined techniques with photo elicitation | 2 (9.1%) |
| Data analysis | Thematic analysis | 8 (36.4%) |
| | Interpretive phenomenological analysis | 4 (18.2%) |
| | Grounded theory-derived content-focused/ systematic analysis. | 5 (22.7%) |
| | Narrative analysis | 2 (9.1%) |
| | Others | 3 (13.6%) |
| Resettlement country | Australia | 10 (45.5%) |
| | USA | 5 (22.7%) |
| | Canada | 3 (13.6%) |
| | Norway | 2 (9.1%) |
| | Germany | 1 (4.6%) |
| | United Kingdom | 1 (4.6%) |

*Note:* Others: Feminist, Miles and Huberman systematic analysis, Explanatory analysis, Feminist- framework approach analysis.

**Table 3.** Quality appraisal of the included studies' qualitative reports

| Author | Q1 | Q2 | Q3 | Q4 | Q5 | Score |
|---|---|---|---|---|---|---|
| DiClemente-Bosco et al. (2024) | Y | Y | Y | Y | Y | 5***** |
| Scott et al. (2022) | Y | Y | Y | Y | Y | 5***** |
| Pittaway and Dantas (2022) | Y | N | Y | Y | Y | 4**** |
| Grupp et al. (2022) | Y | Y | Y | Y | Y | 5***** |
| Goitom and Idemudia (2022) | Y | Y | Y | Y | Y | 5***** |
| Woodgate and Busolo (2021) | Y | Y | Y | Y | Y | 5***** |
| Olukotun et al. (2019) | Y | Y | Y | Y | Y | 5***** |
| Vromans et al. (2018) | Y | Y | Y | Y | Y | 5***** |
| Covington-Ward et al. (2018) | Y | Y | Y | N | Y | 4**** |
| (Ikafa and Hack-Polay, 2018) | Y | Y | Y | Y | Y | 5***** |
| Abraham et al. (2018) | Y | Y | Y | N | N | 3*** |
| Omar et al. (2017) | Y | N | Y | Y | N | 3*** |
| King et al. (2017) | Y | Y | Y | Y | Y | 5***** |
| Joyce and Liamputtong (2017) | Y | Y | Y | Y | Y | 5***** |
| Markova and Sandal (2016) | Y | N | Y | Y | Y | 4**** |
| Betancourt et al. (2015) | Y | Y | Y | Y | Y | 5***** |
| Puvimanasinghe et al. (2014) | Y | N | Y | Y | N | 3*** |
| Shakespeare-Finch and Wickham (2010) | N | Y | Y | Y | Y | 4**** |
| Murray (2010) | Y | Y | Y | Y | Y | 5***** |
| Khawaja et al. (2008) | Y | Y | Y | N | Y | 4**** |
| Schweitzer et al. (2007) | Y | Y | Y | N | Y | 4**** |
| Goodman (2004) | Y | Y | Y | Y | Y | 5***** |

*Note:* Y – Yes; N – No; Q – Appraisal questions for qualitative studies.

sources of support identified included ethnic and home-country community groups, members of specific community associations, regional communities, local neighbours and the host government. Ethnic and home-country communities were repeatedly cited as a significant source of belonging and support. Community networks became a lifeline for many, especially when their nearby family support was unavailable. Female Eritrean individuals noted the "*immense happiness*" brought by reconnecting with familiar faces, underscoring the emotional relief that comes from being surrounded by members of their community (Abraham et al., 2018).

Forcibly displaced people frequently relied on informal conversations, often through phone calls, with other sub-Saharan Africans across the diaspora. These exchanges provided emotional relief, practical advice and a sense of connection. Many people emphasised the importance of support from community associations that are specific to their national and ethnic identities. These associations served as vital hubs where people could reconnect with familiar faces and share their cultural practices. For example, participants became involved in broader African community associations, which provided opportunities to interact with others who had also resettled in the area. Similarly, one Congolese participant noted that being involved in numerous community groups was beneficial to him (Ikafa and Perry, 2023).

The broader regional and host communities were also important sources of support for forcibly displaced people. In regional towns where people were resettled, participants spoke of the local population's and the government's welcoming nature. One Congolese refugee expressed gratitude for the support and effort the community has shown towards them, saying, "I feel like w*e are pretty special to be part of this community. We appreciate the effort that the community has put in for our family ….*"(Joyce and Liamputtong, 2017). Joyce and Liamputtong (2017) also acknowledged the significance of the wider regional community's support for refugees (Joyce and Liamputtong, 2017). Upon arrival, the community's welcoming nature helped the resettled people to feel a sense of belonging in the host environment.

**Family support.** It is reported in nine of the primary studies presented across six codes. For many, the familial support network was integral to coping with displacement and resettlement stressors. This support comes from both immediate and extended families, across home and host countries. Regular communication within families, especially with elders, was seen as a source of reassurance and emotional comfort, happiness and aspirations (Murray, 2010; Betancourt et al., 2015; Joyce and Liamputtong, 2017). For instance, a South Sudanese refugee noted, "*… my grandmother was alive, so if I have anything, I just go and talk to her, and yeah, she comforts me*" (Schweitzer et al., 2007). (Ikafa and Hack-Polay, 2018) also mentioned that most participants coped better with difficulties due to the support they received from their families (Ikafa and Hack-Polay, 2018).

Participants also described the importance of family discussions to address challenges. Talking openly about problems helped family members to share their burdens and offer support to those who need it. One participant who resettled in Australia stated, "*We always talk about [stressful] issues as a family. That's what I think has always helped us since we first arrived in this country. We [also] encourage family members to share their problems.*"(Ikafa and Hack-Polay, 2018) This quote reflects the value of open dialogue and mutual support within familial units.

The support provided by family was not just restricted to adults. Several papers identified how children were also described as a source of motivation and joy that helped them cope with the hardships encountered. Children's academic achievements and future success provided hope and motivation for achieving personal and collective goals "*My kids because my kids are my future. I hope to see them doing very well in their studies, and I hope that next year and the year after, my son will get a high degree and get to university.*" said a South Sudanese parent in Australia (Murray, 2010). This illustrates how family aspirations propel individuals forward, inspiring a sense of purpose and determination to overcome challenges.

**Support from friends.** This sub-theme was synthesised from 10 primary articles (Goodman, 2004; Schweitzer et al., 2007; Khawaja et al., 2008; Murray, 2010; Shakespeare-Finch and Wickham, 2010; Puvimanasinghe et al., 2014; Joyce and Liamputtong, 2017; Ikafa and Hack-Polay, 2018; Olukotun et al., 2019; Scott et al., 2022) across five codes. The quotes emphasise how having diverse friendship networks can help overcome challenges by providing access to essential resources, such as food, money, advice and support, as well as connections to employment opportunities. The friends could be from the home country, from other refugees in the resettlement host country or from making friends with members of the host country. For instance, a Sudanese refugee shared, "*We had to encourage each other and advise each other not to give up, to struggle for a better future*" (Goodman, 2004). The participant emphasises the crucial role of friendship in their life, expressing uncertainty

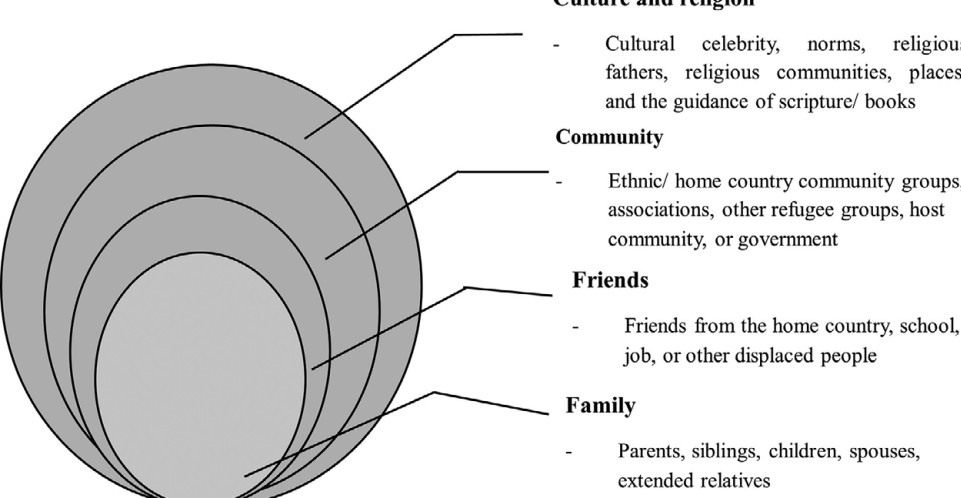

**Culture and religion**

- Cultural celebrity, norms, religious fathers, religious communities, places, and the guidance of scripture/ books

**Community**

- Ethnic/ home country community groups, associations, other refugee groups, host community, or government

**Friends**

- Friends from the home country, school, job, or other displaced people

**Family**

- Parents, siblings, children, spouses, extended relatives

**Figure 2.** Sources of social support for the sub-Saharan African forcibly displaced people in high-income settings.

about how they would have coped without their friends. This sentiment was echoed by others who valued the moral support, advice and mutual encouragement they received from their social circles (Shakespeare-Finch and Wickham, 2010; Ikafa and Hack-Polay, 2018; Olukotun et al., 2019). Friends were also seen as a source of strength, helping to cope with stressors. Furthermore, friendships facilitate resettlement adjustment, such as learning a new language, cultural norms and local practices from their friends (Joyce and Liamputtong, 2017; Ikafa and Hack-Polay, 2018).

**Cultural and religious support.** Culture and religion were frequently identified as key sources of social support for resettled people from sub-Saharan Africa, helping to create continuity in the face of feelings of loss and displacement. It developed from nine primary studies across eight codes. This spanned the domains of culture, the role of personal faith and access to traditional religious and healing practices.

*Cultural identity support.* Engaging in specific cultural rituals and practices, such as storytelling, music, festivals and coffee ceremonies, helps them feel connected to their country of origin. These cultural rituals take various forms across different sub-Saharan African contexts; for instance, storytelling and music composed common elements across many communities with diverse forms of experiences (Cvorovic and Coe, 2022), while coffee ceremonies are a specific tradition among Ethiopian communities (Palmer, 2010). Preserving one's culture between generations has multifaceted benefits, as not only does it keep the culture alive, but it is also a way of connecting to the community and culture in the host country (Puvimanasinghe et al., 2014; Joyce and Liamputtong, 2017; King et al., 2017; Goitom and Idemudia, 2022). As one Congolese participant mentioned, their mother would tell stories and sing songs to help them stay connected to their Congolese roots. "*Mum tells us some stories sometimes and songs…*" (Joyce and Liamputtong, 2017) ensured that both cultural heritages and their new experiences in Australia were passed down to future generations. Ethiopian participants resettled in Toronto, Canada, also described how cultural rituals, such as the Ethiopian coffee ceremony (Goitom and Idemudia, 2022), were practised in their new environment to create a sense of identity and shared experiences. These rituals provided comfort and space for social bonding.

Sub-Saharan Africans also viewed cultural diversity as important and worthy of celebration. It was considered a form of mutual learning and unity among communities. Participants appreciated the host countries' diverse cultures and religions, viewing these differences as opportunities for cultural learning: "*It's really good having people from different cultures and religions, like you, learn a lot from them, and they learn from you.*" (Joyce and Liamputtong, 2017) Cultural diversity also fosters a sense of openness, tolerance and respect for others' perspectives in the resettlement environment.

*Faith-based support.* It was synthesised from four codes and six articles (Schweitzer et al., 2007; Khawaja et al., 2008; Markova and Sandal, 2016; Ikafa and Hack-Polay, 2018; Vromans et al., 2018; Goitom and Idemudia, 2022). Participants in this review described their faith in divine interventions, praying, reading scriptures and attending religious places, such as churches or mosques, as sources of support during the resettlement process. Religious institutions were recognised as more than places of worship; they serve as cultural and community hubs that cultivate resilience and social connectivity. The statement "*they take your hand, guide you, and their network becomes your network, and there is trust because the experience (*i.e., migration and resettlement) builds it*" (Goitom and Idemudia, 2022) underscores the role of religious gatherings in fostering connections with others. The concept of "*mahbär*" (a traditional Ethiopian religious gathering) is a religious practice that also facilitates the establishment of long-term, reliable support networks. It provides consistent social interaction and a sense of belonging, allowing members to rely on one another for various needs. King et al. (2017) also reflected that community mainstreams, such as religious institutions, play a role in the dialogue and interaction of African people (King et al., 2017).

*Religious guidance and healing approaches.* The role of religion extends into mental health support, where religious institutions (churches) act as informal counselling centres for African communities (Betancourt et al., 2015; Markova and Sandal, 2016; Omar et al., 2017; Grupp et al., 2022). For instance, one participant emphasised, "*In Africa, you don't need professional counsellors. You can go to the church and meet a lot of people…*" (Vromans et al., 2018). Scripture readings, such as the Bible and/or the Quran, and guidance from religious leaders were among the known healing

practices used to cure illnesses, and people believed in their potential efficacy.

Furthermore, preferences for religious healing methods vary across demographics. For instance, "*Older Somali men preferred religious healing practices like seeing Sheikhs, while younger Eritrean men leaned towards mainstream services*" (Omar et al., 2017). However, this choice is not always autonomous; sometimes, younger men feel pressure from family members to adhere to such practices, even if they are sceptical of their effectiveness. Several papers found that families may sometimes send their young members back to their home countries for cultural rehabilitation, where they can reconnect with their origins, participate in spiritual rituals and regain a sense of cultural identity. A Somali community representative remarked, "*Especially amongst young men who have drug abuse or mental health issues … there's a growing trend that families send their young ones back home for cultural rehabilitation*" (Omar et al., 2017).

### Theme 2: Reasons for seeking social support

The second core emergent theme across the reviewed studies was the reasons for seeking social support in addressing the challenges associated with resettlement. This theme was synthesised from 11 primary studies across eight codes. In this review, seeking support is understood as part of ongoing relational and community networking, not merely a separate action. Participants described various reasons for seeking social support, which included psychological problems (Goodman, 2004; Betancourt et al., 2015; Markova and Sandal, 2016; Grupp et al., 2022), new social life/networking (Shakespeare-Finch and Wickham, 2010; Joyce and Liamputtong, 2017; Ikafa and Hack-Polay, 2018; Olukotun et al., 2019; Grupp et al., 2022; Scott et al., 2022), disclosure of and discussing problems with others (Abraham et al., 2018; Covington-Ward et al., 2018; Ikafa and Hack-Polay, 2018; Olukotun et al., 2019) and loneliness and a feeling of isolation from family (Goodman, 2004; Markova and Sandal, 2016; Grupp et al., 2022; Scott et al., 2022). Participants stated that their connections with other African and host communities during their resettlement helped them establish new social networks. Conversations with others often helped them manage psychological problems like stress, depression and feelings of anxiety. One Congolese participant reflected this: "*If I get a problem which can cause stress, I go to see my friends from my community, express my feelings to them…*" (Ikafa and Hack-Polay, 2018).

Participants described that isolation from family and friends could worsen mental health, leading to additional challenges. They stressed the importance of forming new social connections and maintaining existing ones to alleviate feelings of isolation. For example, Scott et al. (2022) noted that some individuals felt a connection to their loss of social networks and family connections. They described the importance of engaging in activities and building support networks to distract from worry (Scott et al., 2022).

Furthermore, Somali people described marriage as a solution to mental health stressors that provides companionship to address loneliness and social isolation. In the focus group discussion, a Somali male participant reflected, "*I think that if he marries, he will get somebody he can be with, and then he will not need to feel lonely as he gets somebody to be with, somebody to share with. Happiness and all that.*" Then he added, "*Since he is 27 years old, somebody should find him, somebody to marry! If he marries, he will not be alone anymore. [Most participants nudge at each other] …*"(Markova and Sandal, 2016). This reflects the role of marriage and family establishment in fulfilling social and emotional needs within the community.

### Theme 3: Role of social support

The third emergent theme across the papers concerns the impact of access to social support on refugees' life experiences, which we have identified as the role of social support. The theme was synthesised from the contributions of all the reviewed studies, reflecting the significant thematic weight and consistent prominence of social support across them. It was described across 20 analytic codes. Social support was consistently depicted as essential for providing a sense of community belonging and collective sharing, a source of shared cultural or communal identity and tangible forms of assistance, including practical help, financial support, information and guidance. In addition, social support offered emotional reassurance, relief from distress and depression, opportunities to expand social networks, and facilitated learning, adjustment and resettlement for individuals facing the challenges of displacement.

**Sense of belongingness.** One of the studies documenting the resettlement experiences of Congolese refugees (Joyce and Liamputtong, 2017), noted that participating in cultural or ethnic associations is a core way to connect with others who share similar experiences and backgrounds. This support addresses practical needs and encourages a sense of belonging, emotional comfort and trust within the refugee communities. A participant expressed, "*In Africa, you do not need professional counsellors. You can go to the church and meet a lot of people. People come to you for free; they can talk to you*" (Vromans et al., 2018). This emphasises the African context of support, which relies more on community and religious counselling. Connecting with religious institutions in the new resettlement provides a sense of community connection and linkage with others.

Social contacts provide trust and guidance from experienced individuals, serving as valuable resources for coping and learning (Goitom and Idemudia, 2022). Ethiopians who resettled in Canada described how they regularly gathered to establish religious associations, providing reliable social connections among members. For instance, a female participant stated, "*The mahbär that I belong to has been around for six years. What is valuable about it is that these people form a set group that I can turn to at any moment for anything my family might need. Families come together*" (Goitom and Idemudia, 2022). Somali women in Norway also stated the importance of community engagement in promoting a sense of belonging and connection to their broader social networks: "*Instead of just sitting alone, behind locked doors, it is much better to go to family, friends, or, not least, to neighbours, and talk*" (Markova and Sandal, 2016). It was described as enhancing their well-being and resilience in the resettlement country.

**Reciprocity/collective share and care.** Across the papers, there was discussion of the reciprocal nature of community support. This aspiration is particularly reflected in this quote from a Sierra Leonean refugee resettled in South Australia: "*Giving back to society what society did for us.*" (Puvimanasinghe et al., 2014). Grupp et al. (2022) reported that resettled Somali refugees described a sense of mutual support within their community; a 30-year-old male participant stated, "*I told them that we are Somalis.*" Somalis are like people. They call each other. They wait for each other." (Grupp et al., 2022). Similarly, Goitom and Idemudia (2022) found that Ethiopian refugees resettled in Canada emphasised the importance of faith grounded in core values such as mutual aid, trust, honesty, integrity and commitment. The "*Mahiber*" (association) members described how they shared their concerns with one another and sought advice on topics such as parenting, ways to enjoy

themselves and social interaction. The members also organise and celebrate significant life events collectively, fostering a sense of unity and shared joy, as evidenced by the Ethiopian people (Goitom and Idemudia, 2022).

As many participants described, reciprocity and mutual aid are also central to community patterns. They highlight how community members actively support each other, not only through practical assistance but also through emotional empathy and shared experiences. One Somali mother shared: "*As a mother, we support each other, we give moral support to each other, if someone cries, we try to help them to cry also, we all cry. … We eat together, we laugh together, we cry together.*" (Betancourt et al., 2015) This quote exemplifies the strong bonds of reciprocity and mutual aid that characterise many Somalis and other sub-Saharan Africans, where individuals come together to offer support and solidarity in both joy and hardship. For Eritrean female participants, reciprocity involved finding meaning in past experiences, helping others and seeking support from others during times of hardship (Puvimanasinghe et al., 2014). Forcibly displaced people find strength in solidarity through formal associations, such as the African Association, or informal gatherings within specific linguistic groups. The quote, "*I come from a community where people live in a community, so what troubles me troubles them as well, and they'll be there to offer me encouragement and support*" (Ikafa and Hack-Polay, 2018), epitomises the communal interdependence, wherein hardships are shared, and support is collective.

Practical support. One of the most striking aspects highlighted by the participants is the significance of religious and cultural communities in providing practical support. The role of religious institutions, such as churches, mosques and religious fathers, emerged as pivotal, providing not only spiritual guidance but also tangible aid in the form of financial and practical assistance, as well as emotional solace. A 46-year-old Sudanese man stated, "*You go talk to the father [priest]; this man is good.*" This man helped me with bills, food, and many other things for my kids. Sometimes, he'd give me 20 pounds.*"(Khawaja et al., 2008). Participants also describe how they leveraged community networks to secure employment or access essential resources. For example, one African participant mentioned: "*The next Sunday after the service, a brother in Christ called me and asked if I was looking for a job. I said yes, and he invited me to come to his office the next week.*"(Ikafa and Hack-Polay, 2018).

Relief from distress. The disclosure of problems and seeking social support are highlighted as effective strategies to cope with symptoms of depression and distress (Goodman, 2004; Omar et al., 2017; Vromans et al., 2018; Grupp et al., 2022). Many participants described how communal gatherings facilitated open dialogue and expression of feelings and thoughts, providing a comfortable space. One female participant articulated that it was essential to engage in conversations, as stated, "*…talking to people to cool you down.*" (Covington-Ward et al., 2018) Conversations enable people to release emotional burdens and receive guidance, encouraging them to focus on the present and alleviate their worries. For example, one participant described how others reassured them by saying, "*Don't think about it. Forget those things so that you may live,*"(Goodman, 2004) insisting on how social support mitigates emotional strain and redirects their focus away from distressing thoughts.

Participants noted that being in the company of others provided a valuable distraction from intrusive thoughts and the mental strain that comes from dwelling on negative experiences. One participant remarked that if left alone with their thoughts, "*you can die of thinking*" (Goodman, 2004), but having someone around could play a critical role in diverting their attention away from negative thoughts. Furthermore, after migration, in the absence of close family members, many participants relied on friends or their ethnic community for emotional support. This was especially crucial for individuals who lacked family or intimate partners, such as Somalis living in Norway, who turned to their ethnic community (Markova and Sandal, 2016). The quote, "*If I get a problem which can cause stress, I go to see my friends from my community, express my feelings to them [and] they find ways of assisting me*" (Ikafa and Hack-Polay, 2018), exemplifies how communal networks address emotional distress.

Learning a new life and adaptation. The studies reviewed also describe a process of transition and acculturation, in which it was recognised that, upon arrival, people from refugee backgrounds often cling to their native language and culture. However, as they spend more time in the host country, they tend to adopt the local language and gradually embrace aspects of the host culture (Phinney et al., 2001; Schwartz et al., 2010). Across the reviewed research, sub-Saharan African refugees described actively learning and adapting to new cultural contexts. This process was described in more detail across six of the studies (Shakespeare-Finch and Wickham, 2010; Joyce and Liamputtong, 2017; Omar et al., 2017; Woodgate and Busolo, 2021; Scott et al., 2022; DiClemente-Bosco et al., 2024). For many, the process of adaptation involved learning new social norms, customs and languages, which was facilitated by interaction with others. They described how engaging with people from different backgrounds helped them learn about new ways of living and of integrating into society.

Social support from African refugee members who have been resettled for longer can serve as a bridge for newly resettled individuals to navigate the intricate norms and systems of the host country. One woman participant described support she received from an African refugee woman working in community-led agencies for forcibly displaced people: "*I can just call her, and then she comes to pick me up and take me wherever I want to go*" (DiClemente-Bosco et al., 2024). Participants expressed gratitude for the opportunity to learn from members of other African refugee groups and the host community, who serve as mentors in the journey of cultural integration. These people gain knowledge of language, customs/ foods and everyday practices through friendships with their local communities. For instance, a Congolese participant said, "*Oh yeah, it was important because they [my friends] taught me. I learned a lot from them, like the Australian way. So, they taught me the language, and the food they eat, and the culture, which was really good.*" (Joyce and Liamputtong, 2017) It facilitates the adaptation of people from refugee backgrounds, giving them a sense of belonging in the new environment, with greater confidence and growth.

*Theme 4: Limits of social support*
The other emergent theme identified across the papers relates to the limits of social support, as this review revealed. The theme was synthesised from four analytic codes in the contributions of five studies (Schweitzer et al., 2007; Puvimanasinghe et al., 2014; King et al., 2017; Omar et al., 2017; DiClemente-Bosco et al., 2024). These limits of social support emerged in relation to changes in social norms, preference for individualistic coping, social pressures and geographic dispersal.

Ambivalence or withdrawal from community support structures.    While social support is vital, not all African people reported finding comfort within their ethnic communities. There was discomfort when they were expected to address personal needs in ways that aligned with conventional community practices. Schweitzer et al. (2007) shared that some participants were isolated from their community because they felt misunderstood or pressured to conform to societal norms (Schweitzer et al., 2007). Withdrawing from ethnic and religious networks echoes due to feelings of judgement or pressure to conform or misinterpretations of personal interactions, as one woman participant reflected, "…the *community are calling me and saying he's your husband, he's your boyfriend! I say no. Even if I say that, some people say…you're just lying*" (Vromans et al., 2018). Some individuals, particularly young people, also choose to distance themselves from their communities to rebuild their lives, due to pressure to conform to traditional norms. The narrative from the 19-year-old Eritrean male: "For e*xample, what I did is, I just separated myself from the other people. (…) I started to believe that I had to start a new life. (…) And then I tried to resolve it.*"(Grupp et al., 2022) points to self-driven coping. The more individualistic lifestyle in the host country, where autonomy and independence are prioritised, resulted in weaker communal ties. This cultural shift between communal values and individualistic lifestyles had ramifications for social connectedness, particularly among young people.

Constraints in small or dispersed communities.    In areas with small numbers of ethnic and religious networks, some forcibly displaced people lacked access to familiar support structures or the means to replicate them. King et al. stated that "*many African refugees in Winnipeg, arriving from war-torn regions in small numbers, lack the social and financial resources to organise similar events independently*" (King et al., 2017). This shift led to reduced daily interaction, feelings of isolation and a sense of cultural dislocation. For instance, a woman participant described, "*In Africa, we don't have to call someone … You just go, have fun, all the time. But here, it's [a] difficult time. You close the door… no friend and no visitor, no nothing*" (DiClemente-Bosco et al., 2024). This illustrates how formerly strong communal interactions were weakened by dispersal and resettlement. This social constraint impacted emotional well-being and disrupted communal problem-solving and continuity of familiar communal practices.

## Discussion

Over the past three decades, high levels of political instability and conflict across many sub-Saharan African countries have led to the mass displacement of large populations, with many individuals subsequently resettled in Western countries. The collective and communal nature of traditional life in sub-Saharan Africa, which is central to coping with adversity, is often challenged by forced displacement experiences, including violence and trauma experienced across various stages, as well as resettlement conditions. This disruption is further compounded by intentional dispersal policies, commonly adopted in resettlement contexts, which fragment social networks and limit opportunities to draw on familiar forms of support.

This review synthesised evidence from 22 qualitative studies to examine how forcibly displaced people from sub-Saharan Africa access and use social support during the resettlement period, and the extent to which traditional coping strategies are maintained.

The findings identified a continuity with traditional practices. Across all the studies, participants described relying on multiple sources of social support, including family, friends, community and cultural or religious institutions, to cope with the stressors of displacement. These sources of support were regarded as essential in providing emotional reassurance, practical assistance, cultural preservation, a sense of belonging, and mutual care.

Several studies also described how, in times of crisis, family members may draw upon cultural interventions to support young people who are perceived to have lost their way in the host society. These findings suggest that social systems and networks play a pivotal role in mitigating stress and supporting the well-being of forcibly displaced people from Sub-Saharan Africa. While traditional systems of support remain significant, they are often constrained by the structural and social conditions of the host country. For instance, a dispersal resettlement resulted in weaker communal ties that prevent the rebuilding of the expansive collective networks. Some studies, on the other hand, showed that some people might prefer relying on personal coping rather than social coping, eschewing ethnic and religious networks due to feelings of coercion or pressure to conform.

Moreover, the "limits of social support" theme was less developed than the other themes. This is due to the relatively few quotes and interpretations of social isolation experiences in the included studies. There may be several potential explanations for the limited data on this theme. One is that socially isolated individuals might be less likely to participate in these studies, thereby underrepresenting their perspectives. Another is that participants may have been reluctant to speak negatively about their family and community due to concerns about social desirability or stigma, given the importance of these relationships and norms around harmony and conformity in their culture. of these relationships and norms around harmony and conformity in their culture.

Key elements of the findings reported in this review align with several theoretical frameworks, particularly Conservation Resources (COR) theory, which conceptualises social support, whether practical or emotional, as a key survival strategy in response to resource loss following stressful or traumatic events (Hobfoll, 2011). COR theory posits that individuals and communities are motivated to obtain, retain, and protect material, social, psychological and cultural resources that they value. It further emphasises that resource loss is central to the stress process and tends to exert a disproportionately greater impact than resource gain.

From this perspective, coping strategies such as mobilising community, family and interpersonal social support systems are not only mobilised to replace what has been lost. They also reflect a process of adapting to ongoing threat by building, investing in and sustaining what Hobfoll describes as resource caravans, or clusters of interrelated resources that accumulate over time and reinforce one another. The COR corollaries remind us that those with fewer initial losses, such as young people or small ethnic clusters, may experience intensified impacts from major crises and chronic stressors; they are particularly reliant on these newly formed resource caravans. Meanwhile, we also observed that initial loss often begets future loss- for instance, individuals isolated in resettlement environments struggled to rebuild social ties, leading to further disconnection and stress. In some cases, defensive strategies might have emerged, such as rigid adherence to cultural norms or withdrawal from broader networks as a means of guarding scarce resources (Hobfoll, 2011).

Our findings corroborate the communal dimension of resource emphasised by COR theory. Mutual support, reciprocity and

companionship within social networks indicate how resource conservation operates as a communal rather than an individual process in these contexts. COR theory anticipates this, particularly in its emphasis on resource caravans and caravan passageways, which describe how environmental and social structures shape individuals' access to and maintenance of key resources. The informal networks helped people support each other, as post-migration challenges are communal and shared. In addition, this review highlights the role of cultural identity in rebuilding or maintaining the resources. Forcibly displaced people often rely on cultural and religious practices to regain a sense of self and continuity, suggesting that cultural resources are as vital as material or social ones for resilience (Panter-Brick, 2015; Marie et al., 2018; Fisseha et al., 2025).

The social capital theory, particularly the idea of social capital bonding, is a useful concept for understanding some of the results in this review (Putnam, 2000; Bourdieu, 2011). Social Capital theory posits that networks of relationships generate valuable resources, including trust, reciprocity and mutual support, which individuals can draw on to achieve social and economic outcomes. Our review supports social capital theory by showing that social connections are primarily within refugees' familial, ethnic, religious, cultural or host community networks. While this theory focuses on individual benefits derived from networks, the current review's findings reveal the communal dimension of social capital, in which mutual support and collective care are prioritised over individual gains. Moreover, this review found that the role of cultural institutions, such as religious congregations, as both repositories and facilitators of social capital, reflects their dual purpose as social and spiritual hubs. Religious practices incorporate cultural rituals that serve to soothe the mind, establish a social connection and enhance a sense of connection to their countries of origin (McMichael, 2002; Simich et al., 2003). In many Western contexts, the concept of social is often understood primarily in terms of interpersonal relationships (Yuki, 2003). However, our findings, consistent with anthropological literature, suggest that for many sub-Saharan African communities, the concept of "social" is intrinsically interwoven with spiritual beliefs (Mbiti, 1990). Spiritual rituals often facilitate social networks rooted in a sense of community connectedness, provide mutual assistance and reinforcement of cultural identity. For example, regular attendance at places of worship (church/ mosque services) is a crucial practice that serves as both a spiritual and community hub (Khawaja et al., 2008; Goitom and Idemudia, 2022).

Social capital theory argues that close-knit ties consistently provide benefits. However, our review shows that social bonds are not always beneficial. Some participants distanced themselves from their ethnic communities and then adopted self-driven coping strategies (Grupp et al., 2022) because they might feel pressure to conform to societal norms, and feel uncomfortable when expected to address personal needs aligned with conventional community practices (Schweitzer et al., 2007). Thus, unlike the conventional social capital models, which primarily demonstrate community connections, our review reveals the double-edged nature (both protective and burdensome) of these networks, particularly among forcibly displaced people. Similarly, social capital theory is limited in the context of forced displacement, where networks are disrupted or fragmented. For instance, some participants in smaller or less-established ethnic communities experienced a lack of social and financial resources to mobilise communal events and preserve cultural resources (King et al., 2017), indicating disparities in access to social capital. This emphasises the importance of considering structural barriers that limit the potential of social networks in refugee contexts.

The consistency of findings across the qualitative studies reviewed provides strong evidence that forcibly displaced communities from sub-Saharan Africa perceive their mental well-being as a collective process involving family, friends, community solidarity, culture, faith and societal structures. The consistency of the narratives documented across these studies strongly aligns with a socio-ecological model of mental health (Miller and Rasmussen, 2010). This model locates mental health challenges within multiple socio-ecological levels from individual psychosocial processes to broader family, community and societal systems. It mainly highlighted that post-migration stressors (e.g., discrimination, social isolation and unemployment) often contribute as much, if not more, to mental health outcomes as past trauma (Miller and Rasmussen, 2024).

The findings extend the socio-ecological model, underscoring the dynamic role of forcibly displaced people, families and communities as active agents in shaping their systems. Forcibly displaced people not only interacted with existing structures but also actively transformed them (Ager and Strang, 2008). For instance, participants' interactions with the host community influenced their integration into the broader societal structures (Stewart et al., 2012). The interplay of these structures was dynamic, with cultural continuity within the community level acting as a protective factor of well-being amid systemic adversities (Betancourt and Khan, 2008). This suggests valuing greater attention on the role of collective care and shared cultural practices within the socio-ecological model.

While this was a conceptual breakthrough, our review also found that social support was rarely a central analytic focus in the studies employing this model (Miller and Rasmussen, 2010). In many of the primary studies reviewed, social support was either treated as a background contextual factor or mentioned only peripherally without a deep exploration of how it functioned at different socio-ecological levels. By foregrounding social support in this review, we identified a gap in the application of the socio-ecological framework. Specifically, we found that social support is not merely an environmental context; it is a strategy of adaptation that interacts with identity, cultural continuity, access to services and psychological coping (Simich et al., 2005; Ager and Strang, 2008) and is deeply embedded in individuals' everyday lives. For example, family discussions, community rituals or religious gatherings were shown to shape emotional recovery and cultural preservation actively (Goodman, 2004; Joyce and Liamputtong, 2017). Conversely, some forms of social connectedness created social pressure or exclusion (Schweitzer et al., 2007; Grupp et al., 2022). This dual role of social support in our findings is largely underexplored in prior model applications.

Overall, these gaps in mainstream theoretical models highlight a potentially Western-based conceptualisation of social support as a discrete, individual-level resource that may not adequately reflect the collective and culturally embedded nature of social relationships evident in the studies of sub-Saharan African refugees included in this review. Thus, our findings suggest that existing models should resituate social support as an everyday part of communal adaptation that can be both protective and hindering.

### Practical and policy implications

A key observation that emerges from this review is the extent to which the communities represented by the participants were all active agents in establishing or reestablishing social support systems at the interpersonal, family and community levels, rather than

passive recipients of social support. This underscores the importance of resettlement services and programs in adopting a strength-based approach to developing services for refugees. One that focuses on the capacity of people to cope and flourish despite facing adversity. The roles of informal networks, such as collective care, mutual aid, trust and cultural continuity, are inherent strengths in forcibly displaced people, often overlooked in deficit-focused frameworks (Fazel et al., 2012; Hutchinson and Dorsett, 2012).

Community-driven initiatives that aim to replenish lost resources may be the most effective way to address the mental health challenges of resettlement, which have been extensively documented. The implication for the provision of clinical services is also clear: it is essential to find ways to incorporate cultural and religious practices into the support system. Mental health is not seen as an isolated individual experience, but rather as one with meaning and context within the wider setting of traditional community care and support systems, which should be engaged and supported to ensure culturally congruent care. Wellness on the part of the agency should engage with community processes and support initiatives that strengthen their collective capacity for care and support. Thus, interventions should target multiple levels - individuals, families, communities and structural systems — to promote inclusive well-being. This model of mental health care delivery is sadly not the normative model established in many host countries. The approach, however, is consistent with a growing network of services, such as those established by the linked network of specialist refugee mental health services in Australia that emphasise, as a core element of service delivery to refugee communities, the critical role of community linkage and capacity building (Kaplan, 2020; Aroche and Coello, 2022). These programs can serve as a model for how health and welfare agencies can incorporate strategies such as expanding informal social support networks, facilitating community-building activities and promoting intercultural exchange.

It is crucial to note that the effectiveness of social connections can differ among individuals, which calls for a holistic approach that considers the unique needs and contexts of each person (Seebohm et al., 2013). Additionally, the existence and access to social coping resources may differ across high-income countries, consequently affecting the role of social support for forcibly displaced people (Wood et al., 2021). Nonetheless, factors such as withdrawal from community support structures, normative pressures, limited access to ethnic groups and limited support networks can impede the effectiveness of these networks (Colic-Peisker and Tilbury, 2007; Makwarimba et al., 2010). As a result, the experiences of socially disengaged individuals appear to be underrepresented in the academic literature. This indicates a gap in the current review and underscores the need for future studies to employ methods that can more effectively reach and engage individuals who are less socially connected to community networks.

Furthermore, future research should investigate the longitudinal changes in social support structures and their impact on the mental well-being of forcibly displaced individuals across various phases of displacement and resettlement. Future research should also examine how social ties function for underreported subgroups (LGBTQ+ individuals, people with disabilities).

## Limitations and strengths of the review

This review has several limitations. First, it included only English-language studies, which may have excluded relevant data from studies published in languages other than English. and focuses on adult populations. Second, generalising findings to 'refugees from sub-Saharan Africa' may obscure the significant differences in social support networks across various cultural or ethnic groups. Thirdly, many of the included studies use a cross-sectional design, which captures social support coping at a single point in time. This limits the ability to understand how social support evolves during different stages of resettlement. Since the meta-synthesis is based on secondary analysis of primary studies, it is limited by the scope, depth and quality of the data presented in those studies. Given the lack of group-level cultural comparisons in the studies, our findings could not describe how sources and functions of social support might potentially differ among culturally diverse sub-Saharan African refugees. Furthermore, the findings may not capture the experiences of marginalised sub-groups within people from refugee backgrounds-LGBTQ+ individuals or people with disabilities, who often face unique challenges in accessing social support.

Despite these limitations, this review has several strengths. First, the review employed a rigorous and transparent methodology, adhering to PRISMA guidelines, utilising the MMAT tool for quality appraisal and conducting a three-stage thematic meta-synthesis. The review employed a comprehensive synthesis of diverse quantitative and qualitative studies, providing nuanced insights into the experiences of forcibly displaced people with social support and coping. It enhances the values of qualitative meta-synthesis.

Moreover, our reflexive stance further enriched the analysis by explicitly acknowledging how our cultural backgrounds and field experiences shaped the interpretation of the findings. While we aimed to remain close to the data presented in the included studies, we recognise that our interpretations were influenced by our positionalities. Therefore, we intentionally attended to potential tensions and contradictions in the studies to reflect different sides of social support and to foreground aspects of social support (e.g., faith and spirituality) that have been traditionally neglected in the academic literature. To ensure a balance between our academic stance and personal positionalities while remaining grounded in the data, we iteratively reviewed and revised the themes.

## Conclusion

This review systematically synthesised the sources of social support used by resettled sub-Saharan Africans and their role in coping with stress. Our findings identified diverse sources of social support, including family, friendships, ethnic communities, cultural traditions and religious institutions, as central to coping with the complex adversities of displacement and resettlement. These networks not only provided emotional and practical assistance but also enhanced a sense of belonging, collective resilience and cultural continuity. However, various factors, such as withdrawal from community support structures, normative pressure to conform and the forms of resettlement, can compromise the effectiveness of these networks. The findings support the importance of community-driven approaches that acknowledge the critical role of informal social networks. The services should be multilevel, targeting individual, family and community needs to foster overall well-being. Effective resettlement requires multi-level collaboration between governments, not-for-profit organisations, local councils and previously resettled community groups to develop new support structures. Therefore, it highlights the importance of offering mental health and psychosocial support programs that provide interventions focused on collective forms of care and strengthen social, spiritual and family relationships.

**Open peer review.** To view the open peer review materials for this article, please visit http://doi.org/10.1017/gmh.2026.10150.

**Supplementary material.** The supplementary material for this article can be found at http://doi.org/10.1017/gmh.2026.10150.

**Data availability statement.** The original contributions presented in the study are included in the manuscript.

**Acknowledgements.** During the preparation of this work, the first author(s) used ChatGPT, a language model developed by OpenAI, to support language editing, clarify phrasing, and refine the manuscript's structure. After using this tool/service, the author(s) reviewed and edited the content as needed and take full responsibility for the content of the published article.

**Author contribution.** Conceptualisation: T.K.G., R.W., G.K.; Data Curation: T.K.G., R.W., M.L., Z.S., G.K.; Data Extraction: T.K.G., M.L.; Formal Analysis: T.K.G.; Methodology: T.K.G., R.W., Z.S., G.K.; Software: T.K.G., Supervision: R.W., Z.S., G.K.; Writing–Original Draft, Review and Editing: T.K.G.; Writing–Review and Editing: R.W., M.L., Z.S., G.K. All the authors read and approved the final manuscript.

**Financial support.** This research did not receive any specific grant from funding agencies in the public, commercial, or not-for-profit sectors.

**Competing interests.** The authors declare that there are no competing interests.

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
