## [Reviewer Report]

Dear Editor,

Thank you for the opportunity to review this interesting manuscript.

The manuscript presents a protocol addressing an under-researched yet highly important topic, namely, Social Support Coping Strategies among Sub-Saharan African Forcibly Displaced People, investigated through a systematic review and meta-synthesis. The authors included 22 articles, identifying four key sources of social support among Sub-Saharan African forcibly displaced populations: (1) family, (2) friends, (3) ethnic and community groups, and (4) cultural and religious supports.

The research topic is of considerable importance for the fields of cultural clinical psychology and refugee mental health, as empirical data on non-Western (WEIRD) populations remain scarce. I recommend the publication of this manuscript; however, I would like to offer the following suggestions to further strengthen and refine it:

1. In the title, the authors refer to forcibly displaced people, while in the abstract they use the terms Sub-Saharan Africans and refugees. This makes it somewhat difficult for readers to clearly identify which specific group the authors are referring to, as these designations may include or exclude different populations. It would be helpful to clarify this, as there is a substantial difference between individuals who are officially recognized as refugees and those who hold an insecure or temporary residence status, for example.

2. The introduction could benefit from a slightly clearer structure. At present, it seems to introduce the expected categories of social coping strategies quite early, before fully outlining the broader cultural and contextual background. It might be helpful to first highlight the strong sense of community and the collectivist orientation that characterize many African societies, and then build on this to explain why studying social coping is so relevant. This sequence could make it easier for readers to follow the argument and better understand how forced migration—and, in some cases, collective traumatization—can disrupt or transform these social networks.

3. Page 11 Line 52: I would suggest clarifying the section on cultural identity support. While the authors effectively emphasize a pan-African approach to social support, the opening sentence could unintentionally imply that practices such as storytelling, music, or coffee ceremonies are equally common across all African contexts. Since these traditions vary significantly by region (e.g., coffee ceremonies are specific to certain areas), a brief differentiation would strengthen the section and help avoid potential overgeneralization or cultural stereotyping, especially for readers less familiar with African cultural diversity.

4. Page 11, Line 14: Perhaps a minor remark regarding language and phrasing: is the expression “religious fathers” intentionally chosen as it carries a religious- and gender-specific connotation?

5. On page 19, line 58: I agree with the conclusion that social networks are weakened by resettlement conditions. However, I wonder whether this is truly the only contributing factor. Forced migration itself, along with experiences of violence, trauma, and human rights violations throughout all stages of migration, also profoundly disrupts and damages existing social networks and structures.

6. Page 20, line 31ff: I am unclear why the authors only consider the interpretation that this withdrawal is driven by feelings of coercion or pressure to conform. Social withdrawal could also arise as part of depressive symptoms or other mental health conditions.

7. Intended more as a suggestion or idea: In Western cultures, the concept of “social” is often understood as purely interpersonal or relational. In contrast, research suggests that in some African cultures, the concept of “social” can also encompass a significant spiritual dimension (e.g., relational spirituality, family cohesion through spiritual practices). This provides a valuable opportunity, in the spirit of a cultural formulation, to address the potentially different conceptualizations of the term “social.” Incorporating this perspective could add an important dimension to the manuscript and strengthen the interpretive value of the findings.

---

## [Reviewer Report]

The systematic review provides a useful summary and analysis of a developing research area. The topic of migration is popular and relevant, so a review is a useful contribution.

The paper is overall written to a high standard and makes a valuable contribution to the literature. The paper points theoretically towards explanations for both how refugees demonstrate such resilience, and how some also experience ongoing struggle.

It is commendable that the perspective and positioning of the authors is addressed and explained, as well as explanation of the epistemological stance. The review openly gives recognition to African cultural foundations in making sense of the papers, which can only bring further credibility given the subject.

The introduction is well-structured and provides a sound rationale for the review.

There is a detailed description of the search strategy.

An explanation is provided of the quality appraisal framework, and this has been adhered to in reviewing each of the identified papers.

The analytic method has been briefly but adequately described.

The findings are presented logically and are supported by descriptions and backed up using quotes.

The discussion involves consideration of existing theory, how these could be developed further, and gives recommendation for policy and further research.

Recommendations for improvement:

- Typographical – some of the quotes are italicised and some are not – consistency throughout would ease the readability

- Page 9 – unclear sentence: “In the displaced people’s home country, seven article participants were from four or more different sub-Saharan countries.” I have read this several times and cannot make sense of it.

- Page 14 – theme 3 – could you re-word and explain the first sentence more clearly. “the impact of lived experience in having access to social support, which we have identified as the role of social support.” I wondered if you mean the impact on life experiences of having access to social support?

- Page 18 – line 59-60 add ‘one’ prior to woman

- Page 41 Table 2 – data collection. I think this should add up to 22. Clarify how many studies used face-to-face interviews as I suspect the number should be higher than 1.

Theme 4 – the same quote is used to illustrate both 4.1 and 4.2. It seems this theme is much less developed that the other themes, perhaps due to less detail being present in the studies, with this being less likely to arise through interviews and spoken about by people who agree to participate in research. I think more could be added to the discussion to consider this and what this means for the development of further research. If this area was important enough to warrant a separate theme, what are the implications for people may be struggling with isolation, and who is willing to talk about this to develop understanding further?

Naming of the themes and main conceptual argument – I agreed with the argument that the conceptualisation around social support has given a different focus to how we could understand the importance of more communal aspects of thriving, as opposed to individual strategies that may be more prominent from a westernised perspective. However, I thought there was something missing in the way the themes capture the nuance of how social support could hold significance differently across cultures.

Related to the above, I was expecting in the discussion section a comment or short discussion on how the stance and reflexivity introduced earlier on had influenced the interpretation of findings, or an acknowledgement around how themes developed in context of the cultural awareness that the authors bring. Additionally, an explanation around how social support is defined could be useful, including whether this is a culturally relative framework. For example, it could be that ‘social support’ is an unnecessary term in sub-Saharan Africa, due to the value of community in the culture. This leads to me wondering how does the culture of the academic context influence how themes were developed and named? It also leads me to wonder about the wording of the final sentence of the conclusion, which encourages support to focus on individuals within context, whereas the paper spoke to me more about the value of fostering community and collective care.

Overall, the review is valuable and well-written, offering new conceptual insights. I would like to see the reflexivity around culture more strongly upheld throughout the development of themes and reflected in the discussion.

---

## [Editor Report]

Dear authors,

Thank you for your submission to Global Mental Health. We have now received the required number of reviewers. As you can see, we would like to invite a minor revision of your paper.

Sincerely,

Wietse

---

## [Reviewer Report]

I acknowledge the authors’ efforts to improve the manuscript and consider it suitable for publication. I would like to draw attention to page 21, line 11, as the phrasing appears unclear.

---

## [Reviewer Report]

Thank you for the detailed response to the reviews and taking these on board to amend the paper. I support this paper for publication.

---

## [Editor Report]

Dear authors,

Thank you for submitting a revised version of your paper. Based on the evaluation of the reviewers and my own, we would like to accept the paper, conditional on the change suggested by reviewer 1.

Thank you for submitting your work to our journal!

Warm regards,

Wietse